# A Moderate Reduction of Dietary Crude Protein Provide Comparable Growth Performance and Improve Metabolism via Changing Intestinal Microbiota in Sushan Nursery Pigs

**DOI:** 10.3390/ani11041166

**Published:** 2021-04-19

**Authors:** Riliang Liu, Jianwen He, Xu Ji, Weijiang Zheng, Wen Yao

**Affiliations:** 1Jiangsu Key Laboratory of Gastrointestinal Nutrition and Animal Health, College of Animal Science and Technology, Nanjing Agricultural University, Nanjing 210095, China; t2020082@njau.edu.cn (R.L.); jianwenhe89@163.com (J.H.); jixuchance@163.com (X.J.); zhengweijiang@njau.edu.cn (W.Z.); 2Clinical Research Center, Affiliated Hospital of Shaanxi University of Chinese Medicine, Shaanxi University of Chinese Medicine, Xianyang 712000, China; 3Anhui Province Key Laboratory of Livestock and Poultry Product Safety Engineering, Institute of Animal Science and Veterinary Medicine, Anhui Academy of Agricultural Sciences, Hefei 230031, China; 4Key Lab of Animal Physiology and Biochemistry, Ministry of Agriculture and Rural Affairs of the People’s Republic of China, Nanjing 210095, China

**Keywords:** nursery pigs, protein level, digestibility, blood urea nitrogen, microbiota

## Abstract

**Simple Summary:**

Dietary protein is an important source of amino acids for livestock, while excess protein consumption and an imbalance of amino acids in the diet result in increased feed costs and an increased risk of diarrhea, caused by the proliferation of pathogenic bacteria. Therefore, we investigated whether a diet with a moderate reduction of crude protein levels, supplemented with crystalline amino acids, could alleviate these problems. The results showed that a moderate reduction in the dietary protein level can provide similar growth performance and improve metabolism, due to the improvement of intestinal microbiota, compared with high protein feed in Sushan nursery pigs. These findings present new insights into the role of microorganisms in metabolism, and point to a potential nutrition strategy for indigenous pig production.

**Abstract:**

In this paper, we investigated the effects of a diet with a moderate reduction of dietary crude protein (CP) level, supplemented with five crystalline amino acids (Lys, Met, Thr, Try, and Val), on the growth, metabolism, and fecal microbiota of Sushan nursery pigs. Seventy Sushan nursery pigs with an average body weight of 19.56 ± 0.24 kg were randomly allocated to two experimental dietary treatments: 18% CP (high protein; group HP), and 15% CP (low protein; group LP). We found that the differences in the two diets had no significant effect on the growth performance of Sushan nursery pigs. Nursery pigs on the 15% CP diet showed significantly improved protein, amino acid, and energy utilization. Furthermore, the LP diet cloud optimized the gut microflora composition to some extent. The functional structure of bacterial communities implied improved metabolic capabilities in group LP. Additionally, correlation analysis between fecal microbiota and metabolic profiles confirmed that the increase of beneficial bacterial in the feces was beneficial to the health and metabolism of the nursery pigs. In conclusion, a moderate reduction in the dietary protein level can improve growth and metabolism due to the improvement of intestinal microbiota in Sushan nursery pigs. This finding could provide useful reference data for the application of a different nutrition strategy in indigenous pig production.

## 1. Introduction

Protein is a vital nutrient for all animals, and an inadequate protein intake results in the poor growth and health of pigs in modern pig production [1]. On the other hand, a diet with a high crude protein (CP) level may lead to incomplete digestion and absorption in the gastrointestinal tract (GIT), wasting resources and leading to excessive nitrogen excretion [2,3]. High CP feeds are also associated with a high cost for the raw materials and diarrhea in piglets [4].

The GIT of the pig harbors trillions of microbes, known as the gut microbiota. The gut microbiota interacts with the host by fermenting undigested feed, affecting the host’s intestinal health and nutrient absorption [5,6]. The microbial ecosystem of the GIT is strongly influenced by various factors, however, the intake of the three major nutrients (carbohydrates, protein, and fats) is considered one of the most important [7]. In particular, when piglets are fed high-CP diets, undigested protein contributes to the proliferation of pathogens and results in the production of deleterious metabolites through protein fermentation, which leads to an increase in digestive disorders [8,9].

A potentially effective strategy to alleviate the above problems is to limit dietary CP levels, while balancing the amino acid profile by using crystalline amino acids (CAAs). Previous studies have shown that reducing the dietary CP content, alongside supplementation of CAAs, can enhance the function and health of the GIT, and thus maintain the expected growth performance of pigs [10,11,12]. Other studies, however, have shown that low-CP diets with a balanced amino acid (AA) profile had a negative effect on growth performance and the intestinal health of piglets, especially when dietary CP levels were decreased by more than 4% [13,14]. Moreover, a reduction in dietary CP levels did not necessarily change the composition of the gut microbiota in piglets, although several studies have shown that a moderate reduction of dietary CP levels can help maintain intestinal health by reducing the remount of harmful microbial metabolites [14,15].

Nonetheless, limited data are available concerning the effects of low-protein diets supplemented with CAAs on the composition and functions of the fecal microbiota, and correlations between bacterial and health-related indicators in nursery piglets. In particular, few reports on the optimal dietary CP content of improved local pigs in China are available. The Sushan pig, which genetically is a quarter Chinese Taihu pig, a quarter Danish Landrace pig and half British Yorkshire pig, is a newly developed Chinese breed [16]. Hence, the objective of the present study was to explore the effects of diets with a moderate reduction of dietary CP level, supplemented with five CAAs (Lys, Met, Thr, Trp, and Val) on the growth performance, nutrient digestibility, serum biochemical values, and fecal microbiota of Sushan nursery pigs.

## 2. Materials and Methods

### 2.1. Experimental Design, Animals, and Diets

This study was conducted at the Liuhe Experimental Base in Nanjing, Jiangsu province, China, which is a breeding farm of Sushan pigs (a newly developed Chinese breed). A total of 70 Sushan nursery pigs, with an average body weight of 19.56 ± 0.24 kg, were randomly allocated to two isocaloric diet treatments: 18% CP (high protein; group HP) and 15% CP (low protein; group LP), based on NRC guidelines (2012) [17] (Table 1). In order to meet the requirements of Sushan nursery pigs for essential amino acids (Lys, Met, Thr, Trp, and Val), appropriate levels of these five CAAs were added to the two diets and two diets have the same grams of Lys, Met, Thr, Trp, and Val per MJ of net energy (NE). Each group consisted of five replicate pens. Seven pigs in each pen were reared in an environmentally controlled room, at a temperature of 20–25 °C and relative humidity of 65–70%. Each pen was furnished with two stainless steel feeders and two nipple drinkers, and all of the nursery pigs could access pellet feed and water ad libitum throughout the trial period. The experiment lasted 31 days, which included 3 days for environmental adaptation and 28 days for the feeding trial. The protocol of this study was approved by the Committee of the Animal Research Institute (Certification No. SYXK (Su) 2011-0036), Nanjing Agricultural University, Nanjing, China.

### 2.2. Measurement and Sampling

The body weight of each piglet was recorded on day 1, 14, and 28 to determine the average daily gain (ADG). Feed intake was recorded weekly to calculate the average daily feed intake (ADFI) and the feed/gain (F/G). The fecal and blood samples of the ten nursery pigs from each group were collected (two nursery pigs randomly selected from each pen and a total of 20 of the 70 nursery pigs were selected). Between day 26 and 28, 100 g fecal samples were collected by rectal palpation at 06:30 daily, then were aseptically packed into valve bags and cryopreserved tubes, and frozen at −20 °C or −80 °C for subsequent chemical and microbiota analysis. At 08:00 a.m., 10 mL fasting blood samples were collected from the precaval vein. After centrifugation at 4 °C 3000 r/min for 15 min, the serum was separated and immediately frozen at −20 °C for later analysis.

### 2.3. Chemical Analysis

After drying to a constant weight at 65 °C for 48 h, and grinding and sieving with a 1 mm sieve (40 mesh), the diets and fecal samples were used to analyze the CP, ether extract (EE), crude fiber (CF), and calcium and phosphorus (Ca and P), utilizing Association of Official Analytical Chemists (AOAC) methods (2007) [18]. The gross energy (GE) of the diets and fecal samples was assayed by an automatic adiabatic oxygen bomb calorimeter (TX-6000, Tianxin Instrument, Hebi, China). Acid detergent fiber (ADF) and neural detergent fiber (NDF) of the diets were analyzed using Ankom 200 fiber analyzer (Ankom Technology, Macedon, NY, USA). After dried and acid-hydrolyzed, amino acid analysis of the diets were performed by ino-exchange chromatography on a high-speed analyzer (L-8900; Hitachi, Tokyo, Japan). The digestibility of the dietary components was calculated based on the acid insoluble ash (AIA) that was used as an indigestible marker in the feces [19].

Serum biochemical values, including total protein, albumin, globulin, lipoprotein cholesterol (ALT), aspartate aminotransferase (AST), glucose, urea nitrogen, cholesterol, triglycerides, high-density lipoprotein-cholesterol (HDLC), and low-lipoprotein-cholesterol (LDLC), were evaluated using an automatic biochemical analyzer (Hitachi 7180, Hitachi High-Technologies Co., Tokyo, Japan) and matching commercial kits (Medicalsystem Biotechnology Co., Ltd., Ningbo, China).

The concentration of short-chain fatty acids (SCFAs) in the feces was analyzed by capillary column gas chromatography (GC-14B, Shimadzu, Japan; Capillary Column: 30 m × 0.32 mm × 0.25 μm film thickness; temperature: column 130 °C, injector 180 °C, detector 180 °C) [20].

### 2.4. Fecal Bacterial DNA Isolation, MiSeq Sequencing, and Bioinformatics Analysis

A QIAamp DNA Investigator Kit (QIAGEN, Hilden, Germany) was used to extract total bacterial genomic DNA, following the instructions of the manufacturer. Then, the quantity and quality of each DNA sample were assessed using a NanoDrop spectrophotometer (ND 2000; Thermo, Madison, Wisconsin, MA, USA).

The bacterial 16S rRNA gene V3-V4 hypervariable region was amplified using a universal forward primer, 341F (5′-CCTAYGGGRBGCASCAG-3′), and the following reverse primer, 806R (5′-GGACTCNNGGGTATCTAAT-3′). All PCR reactions were carried out as 30 μL reactions with 15 μL of Phusion ® High-Fidelity PCR Master Mix (New England Biolabs, Inc., Rowley, MA, USA), 0.2 μM of the forward and reverse primers, and about 10 ng of template DNA. Thermal cycling consisted of initial denaturation at 98 °C for 1 min, followed by 30 cycles of denaturation at 98 °C for 10 s, annealing at 50 °C for 30 s, and elongation at 72 °C for 60 s, before the final cycle of 72 °C for 5 min. PCR products were further purified with the GeneJET Gel Extraction Kit (Thermo Fisher Scientific, Waltham, MA, USA). Purified amplicons were pooled in equimolar amounts and paired-end sequenced (2 × 250) on an Illumina Miseq platform at Shanghai Biozeron Biological Technology Co., Ltd. (Shanghai, China), according to the standard protocols [21]. The dates for all samples were deposited into the Sequence Read Archive (http://www.ncbi.nlm.nih.gov/sra/, accessed on 4 March 2021) (Accession Number: SRP309212). Raw fastq files from 16S rRNA Miseq sequencing were first demultiplexed and quality-filtered using QIIME (version 1.17). After merging the paired reads and chimera filtering, operational taxonomic units (OTUs) were clustered within a 97% similarity level using the UPARSE software package (version 7.1) [22], and chimeric sequences were removed using UCHIME [23]. The taxonomies of each OTU were classified using the Ribosomal Database Project (RDP) 16S rRNA classifier against the SILVA (SSU132) 16S rRNA database, with a confidence threshold of 70% [24]. To assess differences in microbial diversity among different samples, we carried out alpha and beta diversity based on the relative abundance of OTUs. Bacterial alpha diversity was estimated by the Mothur program (version 1.21.1) [25], including the ACE, Chao1, and Shannon diversity indices. The differences in bacterial beta diversity were visualized with non-metric multidimensional scaling (NMDS) ordination plots, based on the Bray–Curtis distance metric [26]. Using phyla, genera and OTU with an average relative abundance greater than 1%, 0.1% and 0.1%, respectively, and an occurrence frequency greater than 80%, allowed at least one group to be defined as the dominant bacteria for further analysis [27,28]. In addition, the interactive tree of life (iTOL) was used for tree visualizations, based on the sequences and relative abundance of different OTUs in the HP and LP groups [29].

### 2.5. Statistical Analysis

Power calculations identified a required sample size of ten pigs per treatment group in order to detect an effect size of 1.40 SD for data with 85% power and a type Ⅰ error of 5% by using G*Power Data Analysis [30]. The statistical analysis of growth performance, total tract apparent digestibility, serum biochemical indexes, and SCFAs in the two groups was assessed for normal distribution with the Shapiro–Wilk test. Student’s *t* test (parametric data) or Mann–Whitney *U* test (non-parametric data) was used for two-group comparisions in SPSS 22.0 software (IBM Inc., Chicago, IL, USA). Differences in alpha diversity and relative abundance (phylum, genus and OTU level) of bacterial communities between the two groups were evaluated using the Mann–Whitney *U* test in SPSS 22.0. Data are shown as the means ± standard error of measurement (SEM) for the independent sample *t* test and median for Mann–Whitney *U* test, and difference was considered significant at the value of *p* < 0.05. In the box and whisker plot, the bottom and top edge of the box are the lower and upper quartiles; the upper/lower whisker extends to the highest/lowest value; the band in the middle displays the median. The beta diversity analysis was performed using the unweighted UniFrac distance, to compare the results of the NMDS ordination plot using the community ecology package “vegan” in R (http://www.r-project.org). Structural variation in the microbial community among samples was assessed by permutational multivariate analysis of variance (PERMANOVA). Phylogenetic Investigation of Communities by Reconstrution of Unobserved Species (PICRUSt) was used to predict the function of bacterial communities, and White’s non-parametric *t* test, based on STAMP (version 2.1.3), was applied to detect the differentially abundant Kyoto encyclopedia of genes and genomes (KEGG) pathways. Spearman’s rank correlation analysis was used to assess the associations between significantly changed bacteria and different indexes (including ATTD, serum parameters and SCFAs), and *p* < 0.05 was considered significant for these correlations.

## 3. Results

### 3.1. Growth Performance

As shown in Table 2, the two different protein level diets had no significant effect on the growth performance of Sushan nursery pigs (*p* > 0.05). 

### 3.2. Apparent Total Tract Digestibility

For the apparent total tract digestibility (ATTD, shown in Table 3), in the LP group, the EE (*p* < 0.05) and the GE (*p* = 0.083) digestibility of the nursery pigs increased, but the Ca digestibility (*p* < 0.05) decreased compared with the ATTD of nursery pigs in the HP group.

### 3.3. Serum Parameters

Table 4 shows the effect of dietary protein levels on the serum parameters of nursery pigs. Compared to nursery pigs in the HP group, the concentration of blood urea nitrogen (BUN) was significantly reduced (*p* < 0.01) and the concentration of triglycerides tended to be lower (*p* = 0.087) in the LP group. However, the concentration of total protein, albumin, globulin, ALT, AST, glucose, cholesterol, HDLC, and LDLC showed no significant differences between the two groups (*p* > 0.05).

### 3.4. Fecal Microbial Community 

Figure 1 shows the effect of dietary protein levels on the profiles of fecal microbiota in Sushan nursery pigs. As shown in Figure 1A, the richness (ACE, *p* = 0.071 and Chao1, *p* = 0.076) of fecal microbiota tended to decrease in the LP group. However, the diversity (Shannon and Simpson) shown had no differences (*p* > 0.05) between the two groups (Figure 1B). Furthermore, the NMDS, based on the Bray–Curtis distance metric, showed that the microbial communities in the nursery pigs’ feces were not clearly separated by dietary protein level (*p* > 0.05, Figure 1C).

To further determine the effect of dietary protein levels on the microbial composition of the nursery pigs, the bacterial taxa were subjected to taxonomic composition analysis. Dominant and differential bacterial taxa are shown in Figure 2 and Figure 3, respectively. At the phylum level, Bacteroidetes and Firmicutes were the most predominant phyla in the feces of the Sushan nursery pigs, with a total relative abundance of around 95%. Although there was no statistical difference in Bacteroidetes and Firmicutes abundance between the two groups, the ratio of Firmicutes/Bacteroidetes (F/B) was higher in the LP group than the HP group (1.12 vs. 0.90), because of the increased abundance of Firmicutes (50.13% vs. 45.90%) and the decreased abundance of Bacteroidetes (44.87% vs. 50.13%) in the LP group (Figure 3A). The next most abundant were Proteobacteria and Spirochaeta, and the abundance of Proteobacterial was clearly increased in the LP group (Figure 3A; *p* < 0.05).

The genus-level analysis of the differential genera (Figure 3B) showed that, in the LP group, the relative abundance of seven bacteria genera including *Sphingomonas*, *Terrisporobacter*, *Clostridium sensu stricto 1*, *Bardyrchizobium*, *Achromobacter*, *Halomonas*, and *Caulobacteraceae*, were significantly increased (*p* < 0.05), whereas the relative abundance of *Coprococcus* was significantly decreased (*p* < 0.05).

At the OTU level (Figure 3C), 1017 OTUs were detected in the feces of the two groups of pigs. Among them, 18 of the OTUs, belonging to 13 families, showed significant differences between the two groups (*p* < 0.05). OTUs closely related to Burkholderiaceae, Halomonadaceae, Peptostreptococcaceae, Sphingomonadaceae, Xanthobacteraceae and Clostridiaceae were enriched clearly in the LP group (*p* < 0.05). OTUs closely related to Lachnospiraceae, Muribaculaceae, Prevotellaceae, Rikenellaceae, Ruminococcaceae, Tannerellaceae and Lactobacillaceae, on the other hand, were enriched remarkably in the HP group (*p* < 0.05).

### 3.5. The Predicted Microbial Gene Functions of the Fecal Microbiota

To investigate the effect of the dietary protein level on microbial functions, PICRUSt analysis was performed to predict the relative abundance of metagenomic functions. As shown in Figure 4, 17 of the predicted microbial pathways (level 3) belonging to eight functional categories (level 2) were affected. Compared to the HP group, in the LP group, the relative abundance of ‘Translation’, ‘Translation proteins’, and ‘Transcription’ were decreased and ‘Inorganic Ion Transport and Metabolism’, ‘Lipid Metabolism’, ‘Amino Acid Metabolism’, ‘Neurodegenerative Diseases’, and ‘Xenobiotics Biodegradation and Metabolism’ was increased (*p* < 0.05).

### 3.6. Fecal SCFA Concentrations

The alternations in the bacterial community seen in our two groups could lead to changes in SCFA metabolism; therefore, the SCFA concentrations in the feces of the nursery pigs were determined. The concentrations of acetate were observed to be significantly higher in the LP group (Table 5; *p* < 0.05). However, no effect of dietary protein level was found in the concentrations of propionate, butyrate, isobutyrate, valerate, isovalerate, or total SCFA (Table 5; *p* > 0.05).

### 3.7. Correlations between Fecal Microbiota and Metabolic Profiles

To investigate the contribution of microbial community alteration by dietary protein level to the metabolic status of Sushan nursery pigs, the correlation between fecal microbiota (at the genus and OTU levels) and metabolic profiles (including ATTD, serum parameters and SCFAs) were analyzed (Figure 5).

At the genus level, the relative abundance of *Sphingomonas*, *Bradyrhizobium*, *Achromobacter* and *Caulobacteraceae* were negatively correlated with the ATTD of CF (Figure 5A; *p* < 0.05). The relative abundance of *Coprococcus* was positively correlated with serum concentrations of BUN and ALT and the fecal concentration of propionate, but negatively correlated with the ATTD of GE (Figure 5A; *p* < 0.05). Additionally, the relative abundance of *Clostridiumsensustricto* was negatively correlated with serum BUN and TP and the fecal propionate, while the relative abundance of *Terrisporobacter* was only negatively correlated with serum BUN (Figure 5A; *p* < 0.05).

At the OTU level (Figure 5B), among the OTUs with increased abundance in the HP group, OTU45 (closely related to Rikenellaceae) and OTU293 (Lachnospiraceae) were positively correlated with serum AST, TG, HDL-C, and ATTD of P (*p* < 0.05). OTU106 (Muribaculaceae) and OTU256 (Prevotellaceae) were positively correlated with serum TG (*p* < 0.05). OTU67 (Lachnospiraceae) was positively correlated with serum HDL-C and TG (*p* < 0.05). Conversely, OTU34 (Lactobacillaceae), OTU102 (Tannerellaceae), OTU85 (Ruminococcaceae) and OTU115 (Rikenellaceae) were negatively correlated with serum glucose (*p* < 0.05). OTU67 and OTU160 (closely related to Lachnospiraceae) and OTU85 (Ruminococcaceae) were negatively correlated with fecal acetic acid (*p* < 0.05). On the other hand, among the OTUs with an increased abundance in the LP group, OTU69 (Sphingomonadaceae) and OTU164 (Xanthobacteraceae) were negatively correlated with serum TP and BUN (*p* < 0.05). OTU196 (Burkholderiaceae) was negatively correlated with serum BUN (*p* < 0.05).

## 4. Discussion

Reducing dietary CP level is an effective measure to reduce diarrhea and N excretion in piglets [31,32]. However, the optimal growth performance of pigs depends on adequate AAs in the diet [33]. Previous studies have shown that the first four limiting AAs (Lys, Met, Thr and Trp) should be supplemented when the CP level of a corn-soybean meal diet is reduced, and the fifth (Val) should be considered when the CP level is further reduced [34]. Therefore, in this study, we reduced the dietary CP level by 3% and supplemented these five CAAs (Lys, Met, Thr, Trp and Val) to meet the requirements of nursery pigs. We obtained a comparable growth performance to the 18% CP group, which was consistent with the findings of previous reports [14,15,35]. The comparable ATTD of dietary protein between the LP and HP groups in this study confirmed that the supplementation of these five CAAs (Lys, Met, Thr, Trp, and Val) in a low protein diet can meet the growth requirements of Sushan nursery pigs. Furthermore, the decreased serum BUN in the LP group indicated improved protein utilization efficiency, because of the balanced AAs in the feed [36]. Moreover, serum BUN is highly correlated with the nitrogen concentration in urine, and the decreased serum BUN in the LP group also indicated a decrease in N excretion [37]. Meaningfully, the increase of ‘Amino Acid Metabolism’ related microbial genes, predicted by PICRUSt analysis in the LP group, demonstrated the contribution of the gut microbiota in the improvement of protein and amino acid utilization in Sushan nursery pigs.

The dietary protein level (or the amount of ingested dietary protein) appears to modify both the diversity and the composition of the intestinal microbiota in humans, however, the consequences of such changes to the host physiology and pathophysiology are still poorly understood [5]. Our study also showed that, following the modification of the dietary protein level and the impact that this had on the microbial diversity in the gut of Sushan nursery pigs, there was lower microbial richness (ACE and Chao1) in the fecal microbiota in the LP group, indicating that low-protein diets might decrease the amounts of fecal microbiota in these pigs. With regard to the composition of intestinal microbiota, Firmicutes and Bacteroidetes were the most predominant phyla in the gut, and the ratio of F/B can often reflect the status of microbial homeostasis [38]. An increased ratio of F/B has been widely observed in association with weight gain or obesity; conversely, a decreased ratio has been related to weight loss [39]. In the present study, nursery pigs in the LP group had a higher ratio of F/B and comparable body weight gain compared to HP group with reduced dietary protein levels. Studies on mice fed a high-fat diet showed a significant increase of Proteobacteria abundance [40]. In this study, although the increase of the EE content caused by the increase of corn in the low protein diet was not significant, the abundance of Proteobacteria was still significantly increased in the LP group. However, the increase of Proteobacteria abundance has been shown to be highly correlated with the production of LPS [40], which suggests that wheat, barley, and other feed ingredients with a low EE content may be preferable to corn in the preparation of a low protein diet for pigs. On the other hand, the increase of ‘Lipid Metabolism’-related microbial genes, predicted by PICRUSt analysis in the LP group, showed the contribution of the gut microbiota to the improvement of EE and GE utilization in Sushan nursery pigs. These results are consistent with the results of the higher ATTD of GE and EE in the LP groups.

At the genus level, the relative abundance of *Sphingomonas*, *Terrisporobacter*, *Bardyrchizobium*, *Achromobacter* and *Halomonas* increased in nursery pigs fed an LP diet, and feeding that was reported for the first time in this study. Members of *Sphingomonas* have been reported to be associated with a low concentration of ammonia, and this could be due to their efficient degradation of ammonia and nitrite [41]. Although the concentration of ammonia in feces was not measured in this study, the decrease in the blood urea level in the low protein group might partly reflect that the ammonia in the intestine was used for microbial protein synthesis, rather than urea synthesis. *Terrisporobacter*, a potential acetate-producing bacterium [42], was also increased in the LP group, which is consistent with the significant increase of the fecal acetate concentration in the LP group. Moreover, *Clostridium sensu stricto 1*, which proliferates against the colonization of diarrhea-causing pathogens [43], was increased in the LP group. The association of *Bardyrchizobium*, *Achromobacter*, and *Halomonas* with dietary protein levels requires further exploration. 

At the OTU level, the increase of the relative abundance of OTUs closely related to Peptostreptococcaceae and Clostridiaceae may help maintain gut homeostasis and produce acetic acid, respectively [44,45]. In this study, these two bacteria increased in the LP group, which indicated better microbial homeostasis and a higher capacity for SCFA production in the gut of the LP pigs. The increase of ‘Xenobiotics Biodegradation and Metabolism’-related microbial genes, predicted by PICRUSt analysis, in the LP group also demonstrated the contribution of the microbiota to gut health, through the degrading of toxic compounds such as DON/ZEN (deoxynivalenol and zearalenone) [46,47]. In line with previous studies [48,49,50], the relative abundance of OTUs was closely related to the abundance of Lachnospiraceae, Prevotellaceae, Rikenellaceae and Ruminococcaceae, potential protein-fermenting bacteria, which were remarkably increased in the HP group in this study.

For a deeper understanding of microbial contributions to host physiology and metabolism, the correlations between fecal microbiota at the genera and OTU levels and nursery pigs’ metabolic profiles were analyzed. The OTUs with increased abundance in the HP group contributed to the increase of serum AST, TG, and HDL-C, and to the decrease of serum glucose and fecal acetate. Increased circulating triglycerides are often related to lipid metabolism disorders [51], and increased AST is generally related to hepatic injury. This result indicated that the high protein level in some feeds might harm the hepatic health and lipid metabolism of Sushan nursery pigs. In particular, OTU45 (Rikenellaceae) and OTU293 (Lachnospiraceae) have been reported as biomarkers of hepatic injury and inflammation [52,53]. On the other hand, the OTUs with increased abundance in the LP group were significantly correlated with a decrease of the serum BUN, further confirming the contribution of the gut microbiota to the improvement of protein and amino acid utilization in Sushan nursery pigs.

## 5. Conclusions

In conclusion, Sushan nursery pigs which were given a low protein, balanced amino acid feed (15% CP with five CAAs (Lys, Met, Thr, Trp and Val)) demonstrated a comparable growth performance to pigs given high protein feed (18% CP). The alteration of the gut microbial composition found in the LP group may contribute to an improvement of protein, amino acid, and energy utilization in Sushan nursery pigs.

## Figures and Tables

**Figure 1 animals-11-01166-f001:**
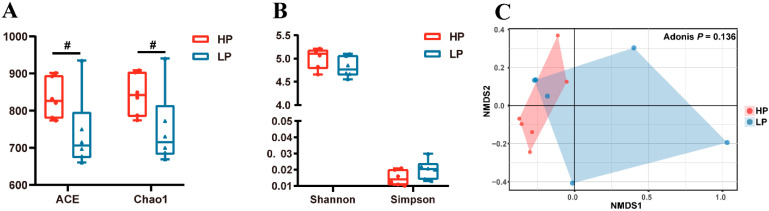
Effects of dietary protein level on the bacterial diversity of feces in nursery pigs. (**A**): A box and whisker display of the species richness of the samples. ACE, abundance-based coverage richness estimate; Chao 1, richness estimate. (**B**): Shannon and Simpson display of the species diversity of samples. (**C**): Nonmetric multidimensional scaling (NMDS) ordination plots of bacterial communities from the feces of nursery pigs. HP = high protein group; LP = low protein group. # indicates 0.05 < *p* < 0.10 (trend).

**Figure 2 animals-11-01166-f002:**
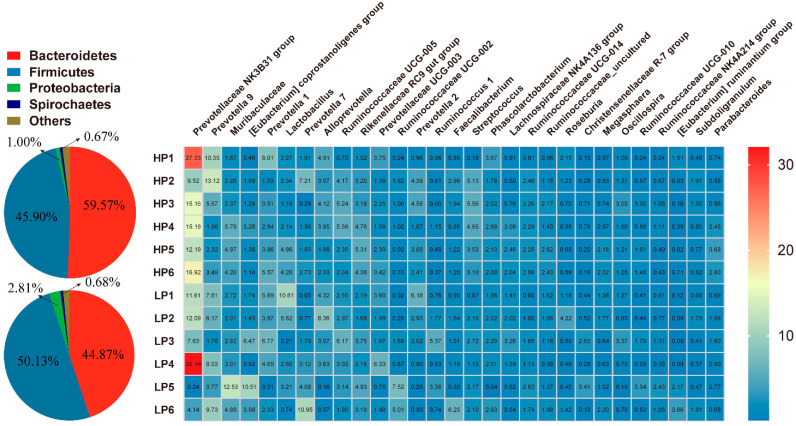
Effects of the dietary protein level on the bacterial composition in the feces of nursery pigs. The pie and heatmap display the dominant phyla, with a > 0.1% proportion and genera for the top 30 in terms of relative abundance in the feces of nursery pigs. HP = high protein group; LP = low protein group.

**Figure 3 animals-11-01166-f003:**
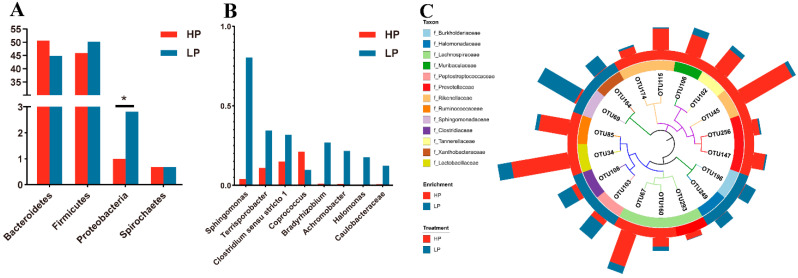
Effects of dietary protein level on the relative abundance of fecal dominant bacteria in nursery pigs. (**A**) Phyla level. (**B**) Genera level. (**C**) The iTOL plots show the phylogenetic trees of the significant dominant OTUs in piglet feces. HP = high protein group; LP = low protein group. * indicates *p* < 0.05.

**Figure 4 animals-11-01166-f004:**
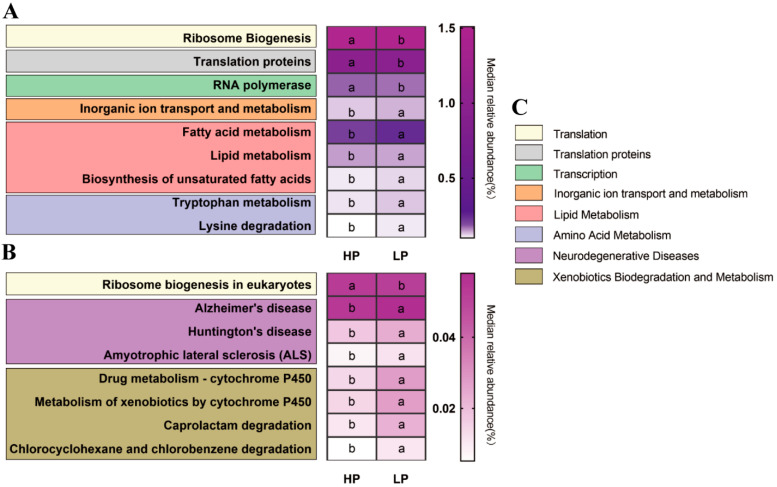
Variations in the KEGG metabolic pathways in functional bacterial communities of feces in nursery pigs. Graph (**A**) shows a relatively high abundance (>0.1%), graph (**B**) shows a relatively low abundance (>0.01%), and graph (**C**) classifies the tertiary metabolic pathways of the secondary metabolic pathways shown in graphs A and B. HP = high protein group; LP = low protein group. “a, b” in the same row indicates *p <* 0.05.

**Figure 5 animals-11-01166-f005:**
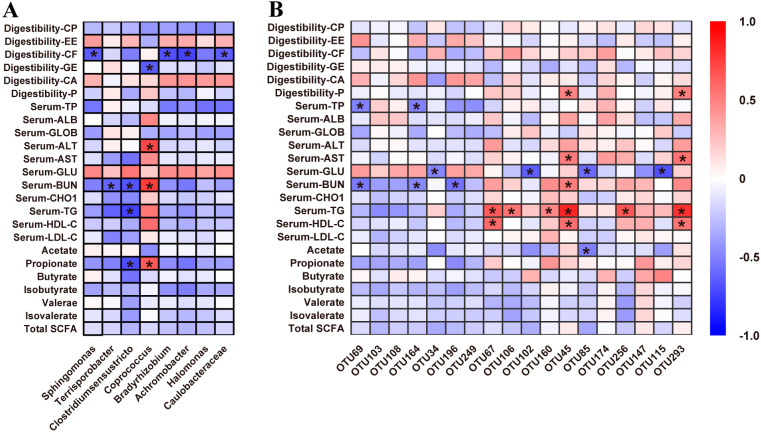
The dietary protein level affects the correlation analysis between differentiated fecal bacteria and different indices of feces in nursery pigs. (**A**) Genus level. (**B**) OTU level. The color indicates the Pearson coefficient distribution. Red represents a positive correlation, while blue represents a negative correlation. * indicates *p* < 0.05.

**Table 1 animals-11-01166-t001:** Ingredient and nutrient composition of the experimental diets (as-fed basis).

Items	Dietary Protein Level ^1^	
	HP	LP
Ingredients, g		
Corn	597.5245	672.7734
Soybean meal	257.9598	160.7904
Wheat shorts	50.00	50.00
Rice bran meal	30.00	30.00
Corn germ meal	30.00	30.00
Dicalcium phosphate	4.00	17.7298
Limestone	10.00	10.00
Salt	5.00	5.00
Premix ^2^	13.275	13.275
L-Lysine	1.3385	4.6934
L-Threonine	0.4087	1.9448
DL-Methionine	0.3907	1.3430
L-Tryptophan	0.1028	0.5796
Valine	0.00	1.8706
Total	1000.00	1000.00
Analyzed nutrient levels, %		
Crude protein	18.89	15.87
Ether extract	2.63	2.80
Acid detergent fiber	11.20	5.91
Neutral detergent fiber	9.43	5.17
Crude fiber	3.79	3.48
Calcium	0.61	0.61
Phosphorus	0.53	0.67
Lysine	1.22	1.23
Methionine	0.28	0.31
Tryptophan	0.21	0.22
Threonine	0.84	0.87
Valine	1.01	1.03
Calculated nutrient levels ^3^, %		
NE ^4^ MJ/kg	10.19	10.21

^1^ HP = high protein group (18% crude protein); LP = low protein group (15% crude protein); ^2^ Premix supplied per kg feed: vitamin A, 10,800 IU; vitamin D3, 4000 IU; vitamin E, 40 IU; vitamin K3, 4 mg; vitamin B1, 6 mg; vitamin B2, 0.05 mg; vitamin B6, 6 mg; vitamin B12, 0.05 mg; nicotinic acid, 50 mg; biotin, 0.2 mg; folic acid, 2 mg; choline, 1000 mg; Fe, 100 mg; Mn, 40 mg; Cu (as copper sulfate) 150 mg; Zn 2000 mg; I 0.5 mg; Se 0.3 mg; acidifier, 4000 mg; anti-mold, 500 mg. ^3^ Values calculated according to the NRC [17]. ^4^ NE, Net energy.

**Table 2 animals-11-01166-t002:** Effect of dietary protein levels on the growth performance of nursery pigs.

Items	Diets		*p*-Value
	HP	LP	
BW, kg			
day 0	19.10 ± 0.35	20.03 ± 0.31	0.152
day 14	26.00 ± 0.50	26.87 ± 0.52	0.230
day 28	33.79 ± 0.70	34.97 ± 0.63	0.158
0–14 day			
ADFI, kg	1.01 ± 0.04	1.03 ± 0.03	0.736
ADG, kg	0.49 ± 0.02	0.49 ± 0.02	0.899
F/G	2.08 ± 0.07	2.11 ± 0.13	0.781
15–28 d			
ADFI, kg	1.26 ± 0.05	1.27 ± 0.05	0.842
ADG, kg	0.53 ± 0.03	0.58 ± 0.03	0.215
F/G	2.29 ± 0.18	2.24 ± 0.12	0.828
0–28 day			
ADFI, kg	1.13 ± 0.04	1.15 ± 0.04	0.765
ADG, kg	0.51 ± 0.02	0.53 ± 0.02	0.359
F/G	2.22 ± 0.06	2.16 ± 0.03	0.421

Data are presented as the mean ± standard error of measurement (SEM, *n* = 10). HP = high protein group; LP = low protein group; BW, body weight; F/G, feed/gain; ADFI, average daily feed intake; ADG, average daily gain; F/G, feed/gain.

**Table 3 animals-11-01166-t003:** Effects of dietary protein levels on total tract apparent digestibility in nursery pigs, %.

Items	Diets		*p*-Value
	HP	LP	
Gross energy	86.75 ± 0.56	88.13 ± 0.49	0.083
Crude protein	80.97 ± 0.94	81.24 ± 1.21	0.865
Ether extract	70.74 ± 1.40	75.45 ± 1.03	0.015
Crude fiber	51.01 ± 5.03	45.01 ± 3.39	0.336
Calcium	48.57 ± 3.16	37.61 ± 3.40	0.030
Phosphorus	33.28 ± 3.47	27.46 ± 3.17	0.233

Data are presented as the mean ± standard error of measurement (SEM, *n* = 10). HP = high protein group; LP = low protein group.

**Table 4 animals-11-01166-t004:** Effects of dietary protein levels on serum biochemical parameters in nursery pigs.

Items	Diets		*p*-Value
	HP	LP	
Total protein, g/L	68.07 ± 1.26	67.83 ± 1.69	0.911
Albumin, g/L	28.58 ± 1.30	26.50 ± 1.04	0.227
Globulin, g/L	39.49 ± 1.55	41.33 ± 2.41	0.529
ALT, U/L	46.67 ± 4.28	45.17 ± 3.61	0.835
AST, U/L	48.83 ± 2.34	49.17 ± 3.34	0.929
Glucose, mmol/L	4.84 ± 0.31	5.33 ± 0.16	0.182
Urea nitrogen, mmol/L	4.54 ± 0.31	2.80 ± 0.17	<0.001
Cholesterol, mmol/L	2.49 ± 0.10	2.58 ± 0.14	0.633
Triglyceride, mmol/L	0.68 ± 0.07	0.53 ± 0.04	0.087
HDLC, mmol/L	0.84 ± 0.06	0.85 ± 0.03	0.928
LDLC, mmol/L	1.21 ± 0.05	1.27 ± 0.08	0.476

Data are presented as the mean ± standard error of measurement (SEM, *n* = 10). HP = high protein group; LP = low protein group; ALT, lipoprotein cholesterol; AST, aspartate aminotransferase; HDLC, high density lipoprotein-cholesterol; LDLC, low lipoprotein-cholesterol.

**Table 5 animals-11-01166-t005:** Effects of dietary protein levels on SCFA concentrations in the feces of nursery pigs, μmol/g.

Items	Diets		*p*-Value
	HP	LP	
Acetate	39.54 ± 2.62	46.72 ± 2.03	0.046
Propionate	18.87 ± 0.77	16.83 ± 1.27	0.191
Butyrate	8.91 ± 0.54	9.01 ± 1.05	0.937
Isobutyrate	2.16 ± 0.42	2.56 ± 1.74	0.558
Valerate	2.90 ± 0.28	3.30 ± 0.59	0.538
Isovalerate	2.59 ± 0.31	2.45 ± 0.34	0.764
Total SCFA	74.50 ± 3.34	80.90 ± 4.79	0.290

Data are presented as the mean ± standard error of measurement (SEM, *n* = 10). HP = high protein group; LP = low protein group; Total SCFA, total short-chain fatty acids.

## Data Availability

Date for all samples were deposited into the Sequence Read Archive (http://www.ncbi.nlm.nih.gov/sra/) (Accession Number: SRP309212).

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
