# Peer review of "A Moderate Reduction of Dietary Crude Protein Provide Comparable Growth Performance and Improve Metabolism via Changing Intestinal Microbiota in Sushan Nursery Pigs"

_animals, 2021, doi:10.3390/ani11041166_

Round 1

Reviewer 1 Report

Dear authors,

I would like to congratulate you on the comprehensive work you have done on the characterisation of the nursery pig intestinal microbiota.

Your results on microbiota characteristics as well as on serum Urea level well document the differences in Nitrogen metabolism of nursery pigs fed diets with different crude protein levels.

I would like to suggest you some minor revisions.

Although the conclusions correctly state that diets did not influence growth performance in pigs, in the summary and in some part of the full text too much emphasis is given to differences in average daily gains (ADG) and F/G between pigs assigned to the 2 dietary treatments.

Summary Lines 21 and 22:  Results show that moderate reduction in dietary protein level can improve growth and metabolism due to the improvement of

Results:

from line 187 -  As shown in Table 2, two protein level diets had no significant effect on the growth performance of Sushan nursery pigs (p > 0.05). However, nursery pigs fed 15% CP diet 187 with balanced AA had greater ADG (0.58 kg/d vs 0.53 kg/d) and lower F/G during phase 15 – 28d than pigs fed 18% CP diet, which indicated nursery pigs gained an additional 50 g of weight per day and consumed comparable feed

from line 338 - The results of present study confirmed these observations: nursery pigs in the LP group had a higher ratio of F/B and higher body weight gain (0.58 kg/d vs 0.53 kg/d).

I personally do not agree with the indications  listed above: because in any case the differences are not statistically significant; because in the period 0-28 the differences between the two groups are minimal.

Line 81 – China with capital C

Line 87 – It could be useful add that the 2 diets have the some grams of Lys per MJ of NE

Line 225 - While  Dominant and differential bacterial taxa were are shown in Fig. 2 and Fig .3, respectively.

Author Response

Point 1: Although the conclusions correctly state that diets did not influence growth performance in pigs, in the summary and in some part of the full text too much emphasis is given to differences in average daily gains (ADG) and F/G between pigs assigned to the 2 dietary treatments.

Summary Lines 21 and 22:  Results show that moderate reduction in dietary protein level can improve growth and metabolism due to the improvement of

Results: from line 187 -  As shown in Table 2, two protein level diets had no significant effect on the growth performance of Sushan nursery pigs (p > 0.05). However, nursery pigs fed 15% CP diet 187 with balanced AA had greater ADG (0.58 kg/d vs 0.53 kg/d) and lower F/G during phase 15 – 28d than pigs fed 18% CP diet, which indicated nursery pigs gained an additional 50 g of weight per day and consumed comparable feed

from line 338 - The results of present study confirmed these observations: nursery pigs in the LP group had a higher ratio of F/B and higher body weight gain (0.58 kg/d vs 0.53 kg/d).

Answer:Dear Reviewer, thank you for your kindly comments and suggestion. We agree, and we have made three changes:

Summary Lines 21-23: Results show that moderate reduction in dietary protein level can provide similar grouth performance and improve metabolism due to the improvement of

Results 205-208, According to the reviewer’s comments, we have deleted this sentence in our manuscript.

Lines 367-369: In the present study, nursery pigs in the LP group had a higher ratio of F/B and comparable body weight gain compared to HP group when reduced ditery protein levels.

Point 2: Line 81 – China with capital C

Answer:Dear Reviewer, thank you for your kindly comments. We have double-checked the spelling problems in our manuscrip and modifiled them (Line 84).

Point 3: Line 87 – It could be useful add that the 2 diets have the some grams of Lys per MJ of NE

Answer:Dear Reviewer, thank you for your kindly comments and suggestion. We have already revised this part according to the suggestions (Line 90-91).

Point 4: Line 225 - While  Dominant and differential bacterial taxa were are shown in Fig. 2 and Fig .3, respectively.

Answer:Dear Reviewer, thank you for your kindly comments. We have polished our expression in the revision (Line 248-249).

Reviewer 2 Report

The objective of this paper was to test the hypothesis that reducing CP level in diets for young pigs would improve growth performance, nutrient digestibility, and indices for intestinal health of Sushan pigs. Seventy pigs were used and were allotted to 2 treatments with 5 replicate pens per treatment (7 pigs per pen). Treatments included a high protein (HP) diet (i.e., 18% CP) and a low protein (LP) diet (i.e., 15% CP). Body weights of pigs were recorded on d 0, 14, and 28 of the experiment. Feed intake was also recorded to calculate ADG, ADFI, and F/G of pigs. Blood samples were collected for serum biomarker analyses. Fecal samples were collected on day 26 to 28 to determine nutrient digestibility in diets and for microbiota analysis. Authors concluded that feeding LP diet had positive effects on nutrient digestibility and growth performance of pigs.

The paper would potentially be a great contribution in the animal industry where the use of antibiotics as growth promoter is banned. However, each of the following comments below should be addressed before the paper can be accepted for publication:

  • It is recommended that authors proofread the manuscript first before submitting. Numerous misspelled words, typographical errors, and grammatical errors were detected. It is also recommended that authors work with a professional editor to highly improve the quality of the manuscript.
  • The analyses conducted for growth performance, nutrient digestibility, etc. only had 5 replicates. Did the authors conduct a power test to accurately determine the number of replicates needed for the experiment? This is an issue that needs to be addressed because few replicates may highly result in false conclusions.
  • The ingredient composition for the LP diet used does not add up to 1000 g/kg. If you sum all ingredients, it adds up to 999.99 g/kg. Please fix.
  • Acid detergent fiber and neutral detergent fiber must be analyzed in the diets. Total dietary fiber analysis is also recommended. Crude fiber analysis is outdated and inaccurate. Amino acids must also be analyzed in the diets (this is highly recommended) since inclusion of crystalline amino acids between the 2 diets are different.
  • Chemical analyses conducted in the diets were not described in the materials and methods. Please include.
  • Were data tested for normality? Were outliers detected? If yes, how? Please describe the procedure.
  • In line 32: What do authors mean with this sentence? Gut microbiota is not something that needs to be ameliorated/alleviated. Please rephrase.
  • Do authors think that 18% CP is considered high in diets for young pigs? Please include evidence.
  • What about diarrhea/fecal scores? This is important since the main objective of reducing CP level in nursery diets is to reduce diarrhea incidence.
  • The procedure described for fecal and blood sampling (i.e., L 105-106) was confusing. Does that mean 2 pigs were used for fecal sampling? Also, 2 pigs were used for blood sampling? Or fecal samples were collected from 1 pig per pen, and a different pig was used for blood sampling?
  • Include the α-level to detect tendency for a significant difference.
  • L 187-190: This statement is incorrect. I believe it is misleading to indicate that pigs fed the LP diet had improved growth performance for d 15-28 if the performance parameters analyzed were not statistically different. Please remove or correct this sentence.
  • L 392-394: Please rephrase. Sentence was a bit confusing.
  • Please change the title. Moderately reducing CP in the diet did not improve growth of Sushan nursery pigs. This is misleading.

Author Response

The paper would potentially be a great contribution in the animal industry where the use of antibiotics as growth promoter is banned. However, each of the following comments below should be addressed before the paper can be accepted for publication:

Point 1: It is recommended that authors proofread the manuscript first before submitting. Numerous misspelled words, typographical errors, and grammatical errors were detected. It is also recommended that authors work with a professional editor to highly improve the quality of the manuscript.

Answer:Dear Reviewer, we thank you for suggesting to polish the language. The manuscript has been carefully checked and revised by professional editor.

Point 2: The analyses conducted for growth performance, nutrient digestibility, etc. only had 5 replicates. Did the authors conduct a power test to accurately determine the number of replicates needed for the experiment? This is an issue that needs to be addressed because few replicates may highly result in false conclusions.

Answer:Dear Reviewer, we are very grateful for your comments. Firstly, we aplogize for the ambiguity caused by our unclear expretion, and we have 10 replicates in each group actually (Line 109-111). We have added the Power calculations by using G*Power Data Analysis in the Materials and Methods (Line 177-179).

Point 3: The ingredient composition for the LP diet used does not add up to 1000 g/kg. If you sum all ingredients, it adds up to 999.99 g/kg. Please fix.

Answer:Dear Reviewer, thank you for your kindly reminding. This problem is caused by rounding, and correct value has been modified (Table1).

Point 4:: Acid detergent fiber and neutral detergent fiber must be analyzed in the diets. Total dietary fiber analysis is also recommended. Crude fiber analysis is outdated and inaccurate. Amino acids must also be analyzed in the diets (this is highly recommended) since inclusion of crystalline amino acids between the 2 diets are different.

Answer:Dear Reviewer, many thanks for the constructive comments. Acid detergent fiber, neutral detergent fiber and amino acids have been analyzed in the diets. The value and materials and methods has been incorporated into the Table 1 and Line 124-127 respectively.

Point 5: Chemical analyses conducted in the diets were not described in the materials and methods. Please include.

Answer:Dear Reviewer, thank you for your kindly reminding. Chemical analyses conducted in the diets was carried out simultaneosly with the chemical analysis of the total tract apparent digestibility. The information has been incorporated into the Materials and methods section (Line 119-124).

Point 6: Were data tested for normality? Were outliers detected? If yes, how? Please describe the procedure.

Answer: Dear Reviewer, thank you for your kindly comments and suggestion. We have already revised this in statistical analysis (Line 177-179) and will expalin these from the following aspects.

Growth performance, total tract apparent digestibility, serum biochemical indexes and SCFAs were assessed for normal distribution with the Shapiro-Wilk test. Student’s t test (parametric data) or Mann-Whitney U test (non-parametric data) was used for two-group comparisions. Differences in alpha diversity and relative abundance (phylum, genus and OTU levels) of bacterial communities between the two groups were evaluated using Mann-Whitney U test in SPSS 22.0.

Point 7: In line 32: What do authors mean with this sentence? Gut microbiota is not something that needs to be ameliorated/alleviated. Please rephrase.

Answer:Dear Reviewer, thank you for your kindly comments. We have already revised this in Line 33-34. Furthermore, the LP diets cloud optimize the gut microflora compostion in some extent.

Point 8: Do authors think that 18% CP is considered high in diets for young pigs? Please include evidence.

Answer:Dear Reviewer, thank you for your kindly comments. In china, a shortage of protein resourses is an important limiting factor to the economic benefit of pig production. China Feed Industry Association issued formula feeds for starter and growing-finishing pigs in 2018. The protein levels of the diets in our study were designed according to this standard, and this standard states that the dietary protein level of pigs at about 20kg should be 15% to 18%. The Sushan pig is a newly developed Chinese breed and have been fed 18% protein level diet in the nursery stage.

Point 9: What about diarrhea/fecal scores? This is important since the main objective of reducing CP level in nursery diets is to reduce diarrhea incidence.

Answer:Dear Reviewer, thank you for your kindly suggestion. We also considered that diarrhea was an important indicator when we did the trial. Unfortunately, there were seven piglets in each pen, and their feces were mixed together and could not be accurately ditinguished.

Point 10: The procedure described for fecal and blood sampling (i.e., L 105-106) was confusing. Does that mean 2 pigs were used for fecal sampling? Also, 2 pigs were used for blood sampling? Or fecal samples were collected from 1 pig per pen, and a different pig was used for blood sampling?

Answer:Dear Reviewer, thank you for your kindly comments. Changed these lines acoording to the suggestions (Line 109-111).

Point 11: Include the α-level to detect tendency for a significant difference.

Answer:Dear Reviewer, thank you for your kindly comments. We think 0.05< P <0.10 is trending and has been modified accordoingly in our manuscrip (Line 244).

Point 12: L 187-190: This statement is incorrect. I believe it is misleading to indicate that pigs fed the LP diet had improved growth performance for d 15-28 if the performance parameters analyzed were not statistically different. Please remove or correct this sentence.

Answer:Dear Reviewer, thank you for your kindly comments and suggestion. This part had been removed (Line 205-208).

Point 13: L 392-394: Please rephrase. Sentence was a bit confusing.

Answer:Dear Reviewer, thank you for your kindly comments. We then polished the text to read: The alteration of the gut microbial composition found in the LP group may contribute to an improvement of protein, amino acid, and energy utilization in Sushan nursery pigs (423-425).

Point 14: Please change the title. Moderately reducing CP in the diet did not improve growth of Sushan nursery pigs. This is misleading.

Answer:Dear Reviewer, thank you for your kindly comments. The title has been changed as requested and we wish reviewer endorse our opinion that such title change does not affect the quality, the conclution, and the significance of our report.

Reviewer 3 Report

The authors carefully analyzed the effect of a moderate reduction in crude protein in the diet that improved growth and metabolism by changing intestinal microbiota in Sushan nursery pigs.

I believe that the article is written correctly from the scientific side.The obtained results were properly described and discussed.My only concern is the number of piglets in the comparative groups – 10. Is this a sufficient number to guarantee the correctness of the statistical inference?Have the authors performed the power analysis of the tests?

I made some minor remarks below:

Line 81: It should be written: China

Line 98, 99: insert space before “mg”

Line 182: “different indexes” – unclear

Line 223: “* indicates p < 0.05.” – this phrase is redundant

Line 336: “in nursery” - was used twice

Line 293-294: OTU106 and OTU256 were positively correlated with serum-TG.

Line 295: OTU67 was positively correlated with HDL-C and Serum TG.

Table 3: I suggest that you give the full variable name (Items) in place of abbreviations.

Figure 1A, 1B: The box and whisker plot – Does the bottom and top edge of the box mean the values of the upper and lower quartiles? I suggest clarifying in M&m.

Figure 1C: What does mean “Adonis” term?

Figure 2. Attach percentages to the chart.

Author Response

Point 1: I believe that the article is written correctly from the scientific side.The obtained results were properly described and discussed.My only concern is the number of piglets in the comparative groups – 10. Is this a sufficient number to guarantee the correctness of the statistical inference?Have the authors performed the power analysis of the tests?

Answer:Dear Reviewer, thank you for your kindly comments and suggestion. We have added the Power calculations by using G*Power Data Analysis in the Materials and Methods (Line 177-179).

Point 2: Line 81: It should be written: China

Answer:Dear Reviewer, thank you for your kindly comments. We have double-checked the spelling problems in our manuscrip and modifiled them (Line 84).

Point 3: Line 98, 99: insert space before “mg”

Answer:Dear Reviewer, thank you for your kindly comments. We have double-checked the spelling problems in our manuscrip and modifiled them (Line 102, 103).

Point 4: Line 182: “different indexes” – unclear

Answer:Dear Reviewer, thank you for your kindly comments. We have polished our expression in the revision (Line 200-201).

Point 5: Line 223: “* indicates < 0.05.” – this phrase is redundant

Answer:Dear Reviewer, thank you for your kindly comments. This part had been removed (Line 244).

Point 6: Line 336: “in nursery” - was used twice

Answer:Dear Reviewer, thank you for your kindly comments. This repeated part had been removed (Line 259).

Point 7: Line 293-294: OTU106 and OTU256 were positively correlated with serum-TG.

Answer:Dear Reviewer, thank you for your kindly comments and suggestion. We have already revised this in line 320-322.

Point 8: Line 295: OTU67 was positively correlated with HDL-C and Serum TG.

Answer:Dear Reviewer, thank you for your kindly comments and suggestion. We have already revised this in line 322-323.

Point 9: Table 3: I suggest that you give the full variable name (Items) in place of abbreviations.

Answer:Dear Reviewer, thank you for your kindly suggestion. We had changed this part according to the suggestions (Table 3).

Point 10: Figure 1A, 1B: The box and whisker plot – Does the bottom and top edge of the box mean the values of the upper and lower quartiles? I suggest clarifying in M&m.

Answer:Dear Reviewer, thank you for your kindly comments and suggestion. In the box and whisker plot, the bottom and top edge of the box are the lower and upper quartiles; the upper/lower whisker extends to the highest/lowest value; the band in the middle displays the median (Line 188-190).

Point 11: Figure 1C: What does mean “Adonis” term?

Answer:Dear Reviewer, thank you for your kindly comments. Adonis is called permutational MANOVA (PERMANOVA) or nonparametric MANOVA.

Point 12: Figure 2. Attach percentages to the chart.

Answer:Dear Reviewer, thank you for your kindly suggestion. We have already revised this part according to the suggestions (Figure 2).

Round 2

Reviewer 2 Report

Please proof-read the manuscript again if possible. Misspelled words are still detected throughout the manuscript.